# CLEAR: Consistent Labeling Enhanced by LLM-driven Automated Re-labeling for Improved Information Retrieval

## Abstract

The performance of information retrieval (IR) systems is heavily influenced by the quality of training data. Manually labeled datasets often contain errors due to subjective biases of annotators, and limitations of retrieval models. To address these challenges, we propose CLEAR, a novel framework that leverages large language models (LLMs) to automatically correct incorrect labels and extract more accurate and true positive documents. CLEAR estimates the reliability of existing annotations using LLMs and rectifies potential labeling errors, thereby improving overall data quality. Furthermore, we conduct a systematic investigation of how utilizing true positive documents affects retrieval model performance. We evaluate CLEAR on several widely-used IR benchmarks, including MS MARCO Passage, MS MARCO Document, Natural Questions, and TriviaQA. Experimental results demonstrate that CLEAR consistently outperforms existing baseline models, validating the effectiveness of the proposed approach.

## 1 Introduction

Natural language processing (NLP) tasks, such as question answering (QA) and information retrieval (IR), typically rely on manually annotated datasets. However, the manual annotation process is inherently susceptible to labeling errors and noise, arising from various factors such as annotator subjectivity, ambiguous annotation guidelines, cognitive biases, and occasional lapses in attention Northcutt et al. (2021); Sheng et al. (2008); Snow et al. (2008); Paullada et al. (2021).

The issue becomes even more pronounced in crowd-sourced annotations involving non-expert workers, where label noise and inconsistencies are substantially more prevalent compared to expert-generated annotations Zhang et al. (2025); Jamison & Gurevych (2015). In tasks such as information retrieval (IR), which require relevance judgments, crowd workers often apply divergent criteria, leading to highly inconsistent labeling Guo et al. (2023). Numerous studies have demonstrated that crowd-sourced annotations are significantly noisier than those produced by trained assessors Chong et al. (2022). Furthermore, several widely used benchmark datasets have been shown to contain a non-negligible number of incorrect labels. Therefore, enhancing dataset quality is essential for the development of robust and reliable natural language processing (NLP) and information retrieval (IR) systems Klie et al. (2023); Agro & Aldarmaki (2023).

Figure 1 illustrates a comparison between a mislabeling example by a human annotator in the MS MARCO dataset Bajaj et al. (2016) and the corrected labeling generated by the proposed CLEAR method. Whereas the human-annotated passage does not explicitly contain the correct answer, the passage labeled by CLEAR clearly provides a precise and direct response to the query.

Incorrect labels can significantly distort the evaluation of retrieval models and impede the training of optimal models. Therefore, ensuring label accuracy is a critical prerequisite for the development of reliable and effective retrieval models. To address this issue, we take inspiration from the human process of labeling documents. In manual annotation, annotators commonly select as positive the document that most clearly provides the correct answer to a given query among those retrieved by a search model. The labeling process can be interpreted as an assessment of how explicitly each document presents the answer to the query. Building on this insight, we propose CLEAR, a novel pipeline that leverages LLMs to efficiently and accurately identify positive documents. CLEAR

Figure 1: An example from the MS MARCO dataset comparing human-annotated and LLM-annotated positive documents for the query "AACN average starting salary of RNs." The ground truth answer is $66,620. The human-annotated document provides general salary ranges for registered nurses but does not explicitly mention the exact answer. In contrast, the LLM-annotated document explicitly states the answer, referencing AACN statistics

is designed to replicate the human labeling process while remaining model-agnostic and broadly applicable across diverse retrieval and LLM configurations.

Recent advances in information retrieval have increasingly emphasized the use of hard negative documents to enhance model performance Zhan et al. (2021); Xiong et al. (2020); Karpukhin et al. (2020); Ren et al. (2021). However, in real-world scenarios, a query is typically associated with multiple relevant documents rather than a single positive instance. This observation underscores the importance of identifying and leveraging a diverse set of positive documents during training Dong et al. (2024); Xu et al. (2019). In this study, we investigate several training strategies designed to effectively incorporate multiple positive documents and conduct systematic experiments to evaluate their impact on retrieval performance. Our findings highlight the critical roles of both the quality and diversity of positive samples, offering practical insights into the development of more robust learning paradigms for information retrieval models.

Our contributions are summarized as follows:

1. We introduce CLEAR, a novel pipeline that leverages LLMs to automatically correct noisy labels in existing information retrieval datasets and construct diverse sets of high quality positive documents. CLEAR emulates the human annotation process to enhance both the accuracy and reliability of training data, and it is designed to be readily applicable across different models and retrieval settings.

2. While prior research has predominantly focused on enhancing retrieval performance through the selection of hard negative documents, we underscore the complementary role of positive document quality and diversity. We propose several training strategies for the effective utilization of multiple positive documents and demonstrate their efficacy through systematic empirical evaluation.

3. We evaluate the effectiveness of CLEAR across a range of widely used benchmark datasets, including MS MARCO Passage, MS MARCO Document, Natural Questions, and TriviaQA. Experimental results show that CLEAR consistently achieves competitive performance relative to strong baselines across all datasets.

## 2 METHOD

Figure 2 presents the overall pipeline of the proposed CLEAR methodology. The CLEAR framework consists of five sequential stages, each of which is described in detail in this section. We particularly emphasize the process of re-labeling Information Retrieval (IR) datasets utilizing LLMs, along with the training strategies designed to effectively leverage the re-labeled data for improving retrieval model performance.

### 2.1 STAGE 1: FINE-TUNING USING HUMAN-LABELED TRAIN DATA

In the first stage, we fine-tune a dense retrieval (DR) model using human-labeled data. Specifically, the DR model is optimized via in-batch negative sampling and the InfoNCE loss Oord et al. (2018);

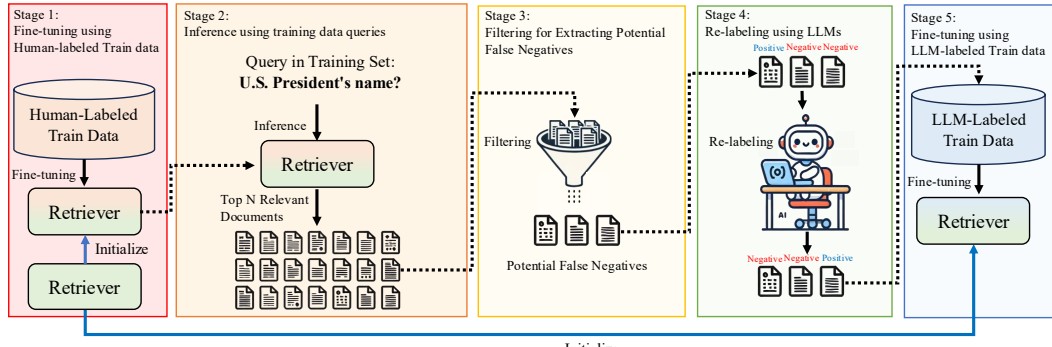

Figure 2: The CLEAR pipeline is designed to improve retriever training through LLM-based re-labeling. The process consists of five stages: (1) fine-tuning an initial retriever using human-labeled training data, (2) running inference on training queries, (3) filtering potential false negatives, (4) LLM-based re-labeling of retrieved documents, and (5) fine-tuning the retriever with the LLM-labeled dataset. The CLEAR framework enhances retrieval quality by correcting label errors and refining training data.

Bertram et al. (2024); Wu et al. (2021). Contrastive learning (CL), a widely adopted framework for training DR models, encourages the model to effectively distinguish positive document pairs from negative ones. The model is trained to minimize the following InfoNCE loss:

$$\mathcal{L}_{\text{CL}} = -\log \left( \frac{\exp(\text{sim}(q, d^+))}{\exp(\text{sim}(q, d^+)) + \sum_{j=1}^{N} \exp(\text{sim}(q, d_j^-))} \right) \quad (1)$$

where $q$ denotes the input query, $d^+$ represents a positive document relevant to the query, $d^-$ indicates a negative document, and $\text{sim}(\cdot, \cdot)$ denotes the dot product between the embeddings of the query and the document.

This initial step establishes the foundation for the subsequent LLM-based automatic re-labeling process, thereby improving both the effectiveness and stability of the CLEAR framework.

## 2.2 STAGE 2: INFERENCE USING TRAINING DATA QUERIES

In the second stage, we perform inference over the entire document collection using the dense retrieval (DR) model fine-tuned in Stage 1. For each query in the training set, the model retrieves the top-$N$ candidate documents with the highest predicted relevance scores.

Let $D$ denote the set of documents retrieved during Stage 2 inference, where $D = \{d_{i,1}, d_{i,2}, \ldots, d_{i,N}\}_{i=1}^{m}$. Here, $m$ is the number of training queries, and for each query $q_i$, the documents $d_{i,j}$ correspond to the top-$N$ candidates retrieved by the DR model according to their similarity scores.

## 2.3 STAGE 3: FILTERING FOR EXTRACTING POTENTIAL FALSE NEGATIVES

Re-labeling all top-$N$ documents retrieved in Stage 2 with LLMs is computationally intensive. To make this tractable, we first select candidate documents that are highly likely to be true positives; we refer to these candidates as *Potential False Negatives* (PFNs). We use a query-specific threshold defined as Threshold $= \tau \cdot s^+$, where $s^+$ is the similarity between the query and its human-labeled positive document. Unless otherwise noted, we set $\tau = 0.95$, which we found most suitable based on a discrete sweep over $\tau \in \{0.70, 0.85, 0.90, 0.95\}$ on the development set (see Appendix A).

The similarity scores between each query and its retrieved documents are defined as $S = \{s_{i,1}, s_{i,2}, \ldots, s_{i,N}\}_{i=1}^{m}$.

Based on the similarity scores $s_{i,j}$, each document $d_{i,j}$ is classified according to the following criteria:

$$d_{i,j} = \begin{cases} \text{Potential False Negative,} & \text{if } s_{i,j} > \text{Threshold} \\ \text{Hard Negative,} & \text{otherwise} \end{cases} \quad (2)$$

$$\text{for } i = 1, 2, \ldots, m, \; j = 1, 2, \ldots, N$$

We define the final set of Potential False Negative (PFN) documents as follows:

$$\text{PFN} = \left\{ d_{i,1}^*, d_{i,2}^*, \ldots, d_{i,k-1}^*, d_{i,k}^+ \right\}_{i=1}^m \tag{3}$$

where $\left\{ d_{i,1}^*, \ldots, d_{i,k-1}^* \right\}$ represents the documents identified as Potential False Negatives, and $d_{i,k}^+$ is the human-labeled positive document for query $i$. By incorporating the Potential False Negatives alongside the human-labeled positive documents, the overall reliability of the training set is enhanced. The filtered PFN documents are subsequently forwarded to the next stage, where they are re-labeled using a large language model (LLM). This selective filtering strategy substantially reduces computational overhead compared to re-labeling all retrieved candidates.

## 2.4 STAGE 4: RE-LABELING USING LLMS

In the fourth stage, we re-label the Potential False Negative documents identified in Stage 3 by leveraging LLMs. We utilize the LLM to generate an answer based on each Potential False Negative document and subsequently compute a confidence score that measures how accurately the LLM generates the correct answer.

Specifically, for each query $q_i$, we construct an input set comprising pairs of PFN documents from Stage 3 and the corresponding answer $a_i$. Formally, this input set is defined as $\mathcal{I} = \left\{ (q_i, d_{i,1}^*, a_i), \ldots, (q_i, d_{i,k}^+, a_i) \right\}_{i=1}^m$.

Each input tuple is provided to the LLM, which computes a document-specific confidence score as $cs = 1 - d(\text{GT}, \; p(y \mid T, q, d))$, where $T$ denotes the prompt template, GT represents a binary vector that indicates the ground-truth answer tokens, and $p(y \mid T, q, d)$ refers to the LLM's predicted probability distribution over the output sequence $y$, conditioned on the prompt $T$, query $q$, and document $d$.

The function $d(\cdot, \cdot)$ computes the distance between the distributions using the length-normalized $L_2$ norm, defined as follows:

$$d(p, q) = \sqrt{\frac{1}{L} \sum_{h=1}^{L} (p_h - q_h)^2} \tag{4}$$

where $p_h$ and $q_h$ represent the $h$-th elements of the probability distributions $p$ and $q$, respectively, and $L$ is the number of tokens in the ground-truth answer. This normalization ensures that the distance measure remains consistent across different sequence lengths.

A higher confidence score indicates that the document allows the LLM to predict the answer with greater accuracy. The complete set of confidence scores is defined as $\mathcal{C} = \left\{ cs_{i,1}, \ldots, cs_{i,k} \right\}_{i=1}^m$.

## 2.5 STAGE 5: FINE-TUNING USING LLM-LABELED TRAIN DATA

In the fifth stage, we propose several re-labeling strategies utilizing the confidence scores $\mathcal{C}$ obtained in Stage 4. Furthermore, we detail the corresponding fine-tuning methodologies designed to effectively exploit the re-labeled samples for improved model performance.

### 2.5.1 FINE-TUNING USING ONLY LLM-LABELED DATA

The first strategy focuses on fine-tuning the model exclusively using positive documents that have been re-labeled by the LLM.

For each query, we select the document with the highest confidence score from the candidate set $\mathcal{C}$ and designate it as the new positive document. Formally, this selection is defined as follows:

$$d_i^{(\text{LLM+})} = \arg\max_k \; cs_{i,k}, \quad \forall i \in \{1, \ldots, m\} \tag{5}$$

where $d_i^{(\text{LLM+})}$ denotes the newly selected positive document, determined according to the confidence scores assigned by the LLM. Subsequently, the model is fine-tuned on these re-labeled documents using the InfoNCE loss function as defined in Equation (1).

### 2.5.2 Augmenting Human-Labeled Data with LLM-Labeled Data

The second strategy entails augmenting human-labeled data with data annotated by an LLM to enhance model performance. The primary motivation for this approach is to address potential omissions or inaccuracies in the human annotations, thereby improving both the quality and the diversity of the dataset.

To this end, we construct an augmented dataset, denoted as $D_{\mathrm{Aug}}$ by combining the human-labeled dataset $D_{\mathrm{Human}}$ with the LLM-labeled dataset $D_{\mathrm{LLM}}$. The human-labeled dataset is formally defined as:

$$D_{\mathrm{Human}} = \left\{ (q_i, d_i^{(\mathrm{Human}+)}, d_i^-) \right\}_{i=1}^{m} \tag{6}$$

The LLM-labeled dataset is defined as:

$$D_{\mathrm{LLM}} = \left\{ (q_i, d_i^{(\mathrm{LLM}+)}, d_i^-) \right\}_{i=1}^{z}, \quad z \le m \tag{7}$$

where $d_i^{(\mathrm{Human}+)}$ and $d_i^{(\mathrm{LLM}+)}$ represent the positive documents selected by the human annotators and the LLM, respectively. To avoid redundancy, any sample in $D_{\mathrm{LLM}}$ that overlaps with the human-labeled positives in $D_{\mathrm{Human}}$ is excluded.

The final augmented dataset is defined as $D_{\mathrm{Aug}} = D_{\mathrm{Human}} \cup D_{\mathrm{LLM}}$, and the model is subsequently fine-tuned on this augmented dataset using the InfoNCE loss function (Equation 1).

### 2.5.3 Joint Training of Human-Labeled and LLM-Labeled Data via Confidence Thresholding

The third strategy is based on the hypothesis that a single query may correspond to multiple positive documents. Under this assumption, all documents whose confidence scores exceed a predefined threshold $\phi$ are regarded as positive examples. Formally, the positive document assignment is defined as follows:

$$d_{i,j} = \begin{cases} \text{Labeled as Positive,} & \text{if } cs_{i,j} > \phi \\ \text{Labeled as Negative,} & \text{otherwise} \end{cases} \tag{8}$$

where $\phi$ denotes the predefined confidence threshold, and $cs_{i,j}$ is the confidence score of the $j$-th document for query $i$. The dataset $D_{\mathrm{LLM}}$, comprising up to $u$ positive documents selected based on the confidence threshold, is formally defined as:

$$D_{\mathrm{LLM}} = \left\{ (q_i, d_{i,1}^{(\mathrm{LLM}+)}, \dots, d_{i,u}^{(\mathrm{LLM}+)}, d_i^-) \right\}_{i=1}^{m} \tag{9}$$

- **Averaging multi-positive (AMP) loss**

We introduce a novel loss function, termed *Averaging Multi-Positive (AMP) Loss*, which is specifically designed to facilitate effective learning from multiple positive documents. AMP Loss promotes balanced optimization by assigning equal importance to all positive samples. Assuming a batch size of 1 for simplicity, the AMP Loss is formally defined as follows:

$$\mathcal{L}_{\mathrm{AMP}} = -\frac{1}{u} \sum_{i=1}^{u} \log \left( \frac{\exp(\mathrm{sim}(q, d_i^+))}{\exp(\mathrm{sim}(q, d_i^+)) + \sum_{j=1}^{N} \exp(\mathrm{sim}(q, d_j^-))} \right)$$

where $u$ is the number of positive documents exceeding the threshold $\phi$, $d_i^+$ represents the $i$-th positive document, and $d_j^-$ denotes a negative document.

- **Confidence-guided multi-positive (CMP) loss**

Although AMP Loss assigns equal weights to all positive samples, this approach may not be optimal because some documents provide much more relevant or clearer answers to the query than others.

To address this limitation, we propose the *Confidence-Guided Multi-Positive (CMP) Loss*, which assigns dynamic weights to positive samples based on their confidence scores predicted by an LLM.

The CMP loss is formally defined as follows:

$$\mathcal{L}_{\text{CMP}} = -\sum_{i=1}^{u} w_i \times \log\left(\frac{\exp(\text{sim}(q, d_i^+))}{\exp(\text{sim}(q, d_i^+)) + \sum_{j=1}^{N} \exp(\text{sim}(q, d_j^-))}\right)$$

where the confidence-based weight $w_i$ is given by:

$$w_i = \frac{\exp(cs_i)}{\sum_{k=1}^{u} \exp(cs_k)} \tag{10}$$

 In this formulation, each positive sample's contribution to the loss is modulated by its associated confidence score, allowing the model to more effectively leverage soft supervision signals generated by the LLM.

## 3 EXPERIMENTAL SETUP

### 3.1 COMPARISON SYSTEMS

To assess the effectiveness of our proposed method, we conduct a comparative evaluation against the following three representative dense retrieval models:

- **DPR :** DPR adopts a dual-encoder architecture that independently encodes queries and documents Karpukhin et al. (2020). The similarity between a query and a document is measured via the dot product of their respective embeddings.

- **CoCondenser:** CoCondenser builds upon the Condenser model by enhancing pretraining with unsupervised learning techniques Gao & Callan (2021). A central contribution is the introduction of *corpus-level contrastive learning*, which strengthens the semantic representations of documents and significantly improves retrieval performance across various benchmarks.

- **DRAGON:** DRAGON advances dense retrieval by employing aggressive data augmentation strategies, including both *query augmentation* and *label augmentation*, to generate a broader diversity of training examples Lin et al. (2023).

- **GTE:** GTE focuses on providing lightweight yet high-performing embedding models for retrieval and semantic similarity Li et al. (2023). GTE emphasizes training efficiency and scalability, using diverse text corpora and optimized objectives to balance performance and computational cost.

- **BGE:** BGE is a family of embedding models developed by BAAI, designed for high-quality text retrieval and semantic representation tasks Xiao et al. (2024). These models leverage contrastive learning with large-scale multilingual corpora and introduce efficient training strategies that make them strong baselines in retrieval benchmarks.

- **LG-ANNA-Embedding:** LG-ANNA-Embedding is a Mistral-7B–based family of general-purpose text embedders designed for both IR and non-IR tasks. The model follows an instruction-based framework that combines in-context prompting with *soft labeling* and *adaptive margin–based hard-negative mining*, enabling strong, scalable semantic representations without architectural changes. Evaluated on MTEB (English, v2), it ranks as the *second-best model overall* on the leaderboard by Borda score Choi et al. (2025).

### 3.2 LLMS USED FOR RE-LABELING

To generate confidence scores and re-label training samples, we leverage a diverse set of LLMs with varying scales and architectural characteristics. Specifically, we utilize LLaMA-3.1-70B, LLaMA-3.1-8B Touvron et al. (2023), EXAONE 3.5-32B Research et al. (2024), Gemma-7B Team et al. (2024), and Qwen 2.5-7B Yang et al. (2024). Among these models, we conduct our experiments using LLaMA-3.1-70B.

## 4 EXPERIMENTS

Table 1 summarizes the retrieval performance of our proposed CLEAR pipeline compared to strong baselines across four benchmark datasets: Natural Questions Kwiatkowski et al. (2019), TriviaQA

| Models | Natural Questions | | TriviaQA | | MS-MARCO (Pas) | | MS-MARCO (Doc) | |
|---|---|---|---|---|---|---|---|---|
| | R@5 | R@20 | R@5 | R@20 | R@5 | R@20 | R@5 | R@20 |
| DPR - Human-only (Pos=1, InfoNCE) | 65.6 | 77.5 | 69.4 | 78.1 | 40.4 | 61.6 | 40.1 | 65.4 |
| DPR - LLM-only (Pos=1, InfoNCE) | 66.1 | 79.3 | 69.5 | 78.4 | 40.5 | 61.7 | 40.0 | 65.4 |
| DPR - Human+LLM Aug (Pos=1, InfoNCE) | 67.1 | 80.2 | 70.6 | 79.9 | 41.4 | 62.9 | 41.1 | 66.2 |
| DPR - Human+LLM Thresh (Pos=N, AMP) | 67.6 | 80.5 | 71.1 | 81.1 | 41.9 | 63.7 | 42.0 | 67.0 |
| DPR - Human+LLM Thresh (Pos=N, CMP) **[CLEAR]** | **68.8** (+3.2%) | **81.1** (+3.6%) | **72.8** (+3.4%) | **81.6** (+3.5%) | **42.4** (+2.0%) | **64.2** (+2.1%) | **42.5** (+2.4%) | **67.5** (+2.5%) |
| CoCondenser - Human-only (Pos=1, InfoNCE) | 72.8 | 80.1 | 73.4 | 80.2 | 45.0 | 68.9 | 43.4 | 71.1 |
| CoCondenser - LLM-only (Pos=1, InfoNCE) | 73.0 | 80.9 | 73.4 | 80.6 | 45.2 | 68.2 | 43.6 | 71.3 |
| CoCondenser - Human+LLM Aug (Pos=1, InfoNCE) | 74.1 | 81.2 | 74.8 | 81.1 | 45.9 | 68.6 | 44.0 | 72.0 |
| CoCondenser - Human+LLM Thresh (Pos=N, AMP) | 74.7 | 82.6 | 75.5 | 82.6 | 46.9 | 69.1 | 44.9 | 72.5 |
| CoCondenser - Human+LLM Thresh (Pos=N, CMP) **[CLEAR]** | **75.7** (+2.9%) | **82.9** (+2.8%) | **76.6** (+3.2%) | **83.3** (+3.1%) | **47.1** (+2.1%) | **70.1** (+1.9%) | **45.5** (+2.1%) | **73.5** (+2.4%) |
| DRAGON - Human-only (Pos=1, InfoNCE) | 71.5 | 81.8 | 73.9 | 82.3 | 53.1 | 74.7 | 48.1 | 74.3 |
| DRAGON - LLM-only (Pos=1, InfoNCE) | 71.9 | 82.1 | 74.1 | 82.4 | 53.7 | 74.9 | 48.6 | 74.9 |
| DRAGON - Human+LLM Aug (Pos=1, InfoNCE) | 72.5 | 82.7 | 75.4 | 84.0 | 54.0 | 75.5 | 49.2 | 75.4 |
| DRAGON - Human+LLM Thresh (Pos=N, AMP) | 72.9 | 83.6 | 75.7 | 84.2 | 54.1 | 76.0 | 49.5 | 75.9 |
| DRAGON - Human+LLM Thresh (Pos=N, CMP) **[CLEAR]** | **73.9** (+2.4%) | **84.4** (+2.6%) | **76.1** (+2.2%) | **84.6** (+2.3%) | **54.9** (+1.8%) | **76.6** (+1.9%) | **50.1** (+2.0%) | **76.6** (+2.3%) |
| GTE - Human-only (Pos=1, InfoNCE) | 72.0 | 82.3 | 74.2 | 83.0 | 53.4 | 75.1 | 48.6 | 75.0 |
| GTE - LLM-only (Pos=1, InfoNCE) | 72.3 | 82.6 | 74.4 | 83.3 | 53.9 | 75.4 | 49.0 | 75.4 |
| GTE - Human+LLM Aug (Pos=1, InfoNCE) | 72.9 | 83.2 | 75.7 | 84.3 | 54.2 | 76.0 | 49.6 | 76.0 |
| GTE - Human+LLM Thresh (Pos=N, AMP) | 73.5 | 84.0 | 76.0 | 84.6 | 54.6 | 76.6 | 49.9 | 76.5 |
| GTE - Human+LLM Thresh (Pos=N, CMP) **[CLEAR]** | **74.4** (+2.4%) | **85.0** (+2.7%) | **76.8** (+2.6%) | **85.3** (+2.3%) | **55.5** (+2.1%) | **77.4** (+2.3%) | **50.8** (+2.2%) | **77.3** (+2.3%) |
| BGE - Human-only (Pos=1, InfoNCE) | 72.4 | 82.6 | 74.6 | 83.3 | 53.7 | 75.4 | 48.9 | 75.3 |
| BGE - LLM-only (Pos=1, InfoNCE) | 72.7 | 82.9 | 74.8 | 83.6 | 54.0 | 75.7 | 49.2 | 75.7 |
| BGE - Human+LLM Aug (Pos=1, InfoNCE) | 73.3 | 83.6 | 75.9 | 84.6 | 54.5 | 76.3 | 49.8 | 76.3 |
| BGE - Human+LLM Thresh (Pos=N, AMP) | 73.9 | 84.4 | 76.3 | 84.9 | 55.0 | 76.9 | 50.1 | 76.8 |
| BGE - Human+LLM Thresh (Pos=N, CMP) **[CLEAR]** | **74.6** (+2.2%) | **85.0** (+2.4%) | **77.0** (+2.4%) | **85.5** (+2.2%) | **55.8** (+2.1%) | **77.7** (+2.3%) | **51.0** (+2.1%) | **77.3** (+2.0%) |
| LG-ANNA-Embedding - Human-only (Pos=1, InfoNCE) | 72.8 | 82.9 | 75.0 | 83.6 | 54.0 | 75.6 | 49.2 | 75.6 |
| LG-ANNA-Embedding - LLM-only (Pos=1, InfoNCE) | 73.1 | 83.2 | 75.2 | 83.9 | 54.3 | 75.9 | 49.5 | 75.9 |
| LG-ANNA-Embedding - Human+LLM Aug (Pos=1, InfoNCE) | 73.8 | 83.9 | 76.3 | 85.0 | 54.9 | 76.6 | 50.1 | 76.6 |
| LG-ANNA-Embedding - Human+LLM Thresh (Pos=N, AMP) | 74.4 | 84.8 | 76.8 | 85.3 | 55.5 | 77.3 | 50.8 | 77.1 |
| LG-ANNA-Embedding - Human+LLM Thresh (Pos=N, CMP) **[CLEAR]** | **75.2** (+2.4%) | **85.6** (+2.7%) | **77.6** (+2.6%) | **86.1** (+2.5%) | **56.4** (+2.4%) | **78.4** (+2.8%) | **51.7** (+2.5%) | **78.0** (+2.4%) |

Table 1: Performance comparison of various retrieval models across four datasets, evaluated using Recall@5 and Recall@20 metrics. Models are trained with InfoNCE Loss (InfoNCE), Averaging Multi-Positive Loss (AMP), and Confidence-guided Multi-Positive Loss (CMP). Our proposed method, CLEAR, which leverages LLM-generated positives selected based on confidence scores, consistently outperforms the baselines. Percentage improvements over the baselines are reported in parentheses.

Joshi et al. (2017), and MS MARCO Bajaj et al. (2016). All datasets used in our study are in English and primarily cover web-based passages and open-domain questions. MS MARCO consists of real anonymized Bing queries and passages retrieved from web documents. Natural Questions consists of real, anonymized queries issued to the Google search engine, paired with Wikipedia articles retrieved at the time of the query. Our re-labeled dataset inherits these properties. For the Natural Questions and TriviaQA datasets, we use the same train/dev/test splits as provided in the original benchmark releases Karpukhin et al. (2020). For MS MARCO, we use the publicly available dataset without any modification. Retrieval performance is measured using Recall@5 and Recall@20 metrics.

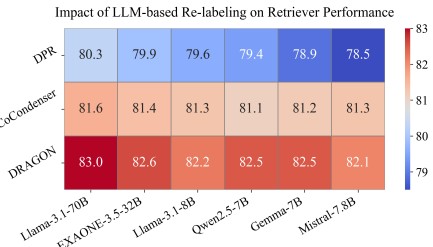

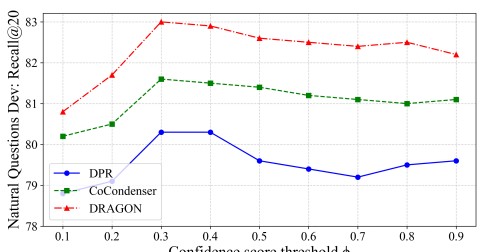

Figure 3: Retriever performance across different LLMs on the Natural Questions dataset, measured by Recall@20. The heatmap compares the retrieval effectiveness of three retrievers when paired with various LLMs, including Llama-3.1-70B, EXAONE-3.5-32B, and others. Higher recall scores are indicated in red, while lower scores are in blue.

Figure 4: Impact of confidence score threshold $\phi$ on Recall@20 for the Natural Questions development set. The plot compares the performance of three models as the confidence threshold varies from 0.1 to 0.9.

### 4.1 EFFECT OF LLM-BASED LABELING (LLM-ONLY)

Training on *LLM-only* labels yields small but consistent gains across all six retrievers (DPR, Co-Condenser, DRAGON, GTE, BGE, LG-ANNA-Embedding). While a few metrics show marginal regressions, the overall trend is positive, indicating that LLMs both correct a fraction of noisy human labels and surface answer-bearing documents missed by annotators. These improvements appear without any architectural changes, underscoring that the *quality of supervision*—rather than model capacity alone—is the main driver. A label-level audit further shows that roughly $10\%$ of documents are re-labeled by the LLMs, whereas about $90\%$ match human judgments.

### 4.2 IMPACT OF AUGMENTING DATA WITH BOTH HUMAN AND LLM LABELS (HUMAN+LLM)

Augmenting human-annotated data with labels generated by LLMs consistently improves retrieval performance. This finding suggests that human-labeled and LLM-labeled documents serve complementary functions, jointly contributing to enhanced retrieval effectiveness.In many cases, both human-annotated and LLM-labeled documents can be considered valid positive examples, reflecting the multiplicity of relevance judgments.These findings underscore the value of combining human and LLM supervision to construct richer and more semantically diverse training signals, ultimately leading to more robust retrieval models.

### 4.3 EFFECTIVENESS OF MULTI-POSITIVE TRAINING (JOINT TRAINING, AMP LOSS)

Training with multiple positive documents consistently outperforms training with a single positive document. These findings indicate that leveraging multiple positive examples facilitates more stable and robust model learning. We hypothesize that this improvement stems from the increased diversity and coverage provided by multi-positive supervision. In contrast to single-positive training, where the model is optimized to match a narrow view of relevance, multi-positive training exposes the model to a wider semantic spectrum of valid answers. This helps the model generalize better to unseen queries by reducing overfitting to a limited set of lexical or structural patterns. Additionally, averaging over multiple positives during loss computation smooths the learning signal and mitigates the influence of outlier examples, further contributing to optimization stability and performance robustness.

### 4.4 EFFECTIVENESS OF CONFIDENCE-GUIDED WEIGHTING (JOINT TRAINING, CMP LOSS)

In multi-positive training, uniformly assigning weights to all positive documents may not always yield optimal performance because not all positives contribute equally to a given query. To address this, we employ a confidence-guided weighting strategy that dynamically adjusts each positive's contribution based on LLM-provided confidence scores. This is particularly beneficial when some

LLM-labeled positives are only weakly relevant or noisy: by down-weighting low-confidence examples, the model avoids overfitting to uncertain supervision signals and allows high-confidence positives to exert greater influence during training. Empirically, across all retrievers we tested (DPR, CoCondenser, DRAGON, GTE, BGE, and LG-ANNA-Embedding), CMP loss yielded the best performance among the compared objectives (InfoNCE, AMP, CMP).

## 4.5 COMPARATIVE ANALYSIS OF RETRIEVER PERFORMANCE WITH VARIOUS LLM LABELERS

Figure 3 illustrates the impact of LLM-based re-labeling on the training of retrieval models. In the proposed framework, an LLM receives a query and a document as input and generates a binary judgment indicating whether the document contains the correct answer. Larger-parameter LLMs possess greater parametric knowledge, enabling them to produce more accurate and reliable labels.

*Experimental setup.* For this analysis, we restrict the retriever backbone to one of three representative models—DPR, CoCondenser, DRAGON—and train each exclusively with labels produced by a given LLM, while holding all other configurations fixed.

*Results.* Retrieval performance improves as the labeler's parameter scale increases. Notably, DRAGON achieves the highest Recall@20 when trained with labels from LLaMA-3.1-70B, closely followed by EXAONE-3.5-32B. In contrast, relatively smaller labelers—such as LLaMA-3.1-8B, Qwen2.5-7B, and Gemma-7B—yield comparable outcomes, whereas Mistral-7B consistently produces the lowest Recall@20 across the three retrievers. This suggests that lower-quality answers from smaller LLMs can degrade label quality and, in turn, negatively affect downstream training.

The same monotonic trend with labeler quality is observed across all three retrievers (DPR, CoCondenser, DRAGON), underscoring the importance of selecting a sufficiently large LLM for re-labeling: high-quality supervision from high-capacity models substantially enhances retrieval effectiveness.

## 4.6 IMPACT OF CONFIDENCE SCORE THRESHOLD ON RETRIEVAL PERFORMANCE

Figure 4 presents the impact of the confidence score threshold ($\phi$) on retrieval performance. The figure compares Recall@20 across three models—DPR, CoCondenser, and DRAGON—under varying threshold values, highlighting how filtering based on LLM-generated confidence scores affects retrieval quality.

Overall, increasing the confidence score threshold leads to a decrease in Recall@20. This trend indicates that overly aggressive filtering based on high confidence scores may inadvertently exclude valuable positive samples, thereby impairing retrieval effectiveness.

The highest performance is observed at $\phi = 0.3$, suggesting that removing low-confidence, potentially noisy positive samples can contribute to improved model training. At $\phi = 0.3$, an average of 3.5 positive documents are retained per query. These results suggest that maintaining a lower confidence threshold, which allows for a greater diversity of positive documents during training, can further enhance retrieval performance.

## 5 CONCLUSION

In this work, we propose CLEAR, a novel pipeline that improves the quality of IR training datasets via LLM-based re-labeling. By correcting noisy labels and identifying diverse, high-quality positives, CLEAR enhances both the accuracy and coverage of supervision.

Experiments on four benchmark datasets show that CLEAR consistently improves retrieval performance across multiple retrievers. We also demonstrate that confidence-guided weighting in multi-positive training stabilizes optimization and enhances generalization.

These results underscore the value of LLMs as effective tools for constructing reliable IR datasets and motivate future research on automated label refinement and soft-supervision in retrieval tasks.

## 6 ETHICAL CONSIDERATIONS

**Data Source Transparency**  We use only publicly available datasets—MS MARCO, Natural Questions, TriviaQA, and others—which were released for academic use and contain no personally identifying information. No additional human data was collected or annotated.

**Bias and Fairness Concerns**  While CLEAR aims to improve label quality, it inherits potential biases from both the original human annotations and the LLM used for re-labeling. For example, LLM-generated answers may reinforce patterns present in web-scale pretraining data, leading to unintentional biases in re-labeled datasets.

**Responsible Use**  Our re-labeled data and pipeline are intended strictly for academic research. Practitioners adopting CLEAR should be cautious about unintended consequences of relying on LLM-generated pseudo-labels, especially in sensitive application domains. Future work should explore mechanisms to verify or calibrate LLM-generated outputs for better safety and transparency.

## 7 REPRODUCIBILITY STATEMENT

We release anonymized code, configuration files, and logs (URL omitted for review). All results are averaged over three random seeds $\{13, 21, 42\}$ and reported as mean $\pm$ std. Experiments are run on a single NVIDIA A100-SXM4-40GB GPU with Python v3.8, PyTorch v1.9.0, CUDA v11.1, and Transformers v4.22.1.

For fair comparison, we keep the retrieval depth, input lengths, and the number of hard negatives fixed (one per query) across all baselines and our method. We adopt each model's recommended batch size, and we use the officially recommended hyperparameters (optimizer, scheduler, learning rate, weight decay, dropout, gradient clipping, precision, and maximum sequence lengths).

During training we select the checkpoint with the highest dev Recall@20 using early stopping (patience=3) and then evaluate once on the test set. LLM-based labeling uses fixed decoding settings (temperature=0.0, top-p=1.0, max_new_tokens=32) and released prompt templates.

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

## A  IMPACT OF FILTERING THRESHOLD ON RETRIEVAL PERFORMANCE

| Model : DPR | Llama-3.1-70B | Llama-3.1-8B |
|---|---|---|
| Threshold : 0.95 | 66.1 | **65.4** |
| Threshold : 0.90 | 66.3 | 65.1 |
| Threshold : 0.85 | 66.3 | 65.0 |
| Threshold : 0.80 | **66.5** | 64.7 |
| Threshold : 0.75 | 66.4 | 64.5 |
| Threshold : 0.70 | 66.2 | 64.6 |

Table 2: Recall@5 for DPR LLM-only under different thresholds on the NQ dataset

In Stage 3, we apply similarity-based filtering to identify potential positive candidates via a thresholding strategy. Because re-labeling every retrieved document with LLMs is computationally expensive and often unnecessary, we first narrow the candidate pool to reduce labeling overhead. Concretely, we compute similarity between each candidate and the human-labeled positive(s) for the same query and retain only those above a threshold $\tau$; LLM-based re-labeling is then applied solely to this filtered subset.

To validate this design, we conducted a sweep over threshold values (e.g., $\tau \in \{0.70, 0.75, 0.80, 0.85, 0.90, 0.95\}$) and found that $\tau = 0.95$ offered a strong balance between recall of potential positives and exclusion of semantically close hard negatives. We also observed that higher thresholds tend to yield slight performance gains, particularly when stronger LLMs are used for re-labeling; larger-parameter models exhibit better semantic discrimination, enabling more accurate re-labeling even as the filter is relaxed and a broader set of candidates is considered.

## B  HUMAN VS. LLM: KEY DIFFERENCES IN IR RELEVANCE LABELING

This appendix summarizes practical differences between human- and LLM-labeled data for information retrieval (IR), focusing on quality/nuance, cost/speed, and error profiles in *relevance judgments*.

### 1. Quality and Nuance

**Human.** Human assessors infer user intent (explicit/implicit), handle ambiguous or multi-intent queries, and account for temporal freshness, credibility, redundancy, and graded relevance. They can judge collection-specific policies (e.g., passage vs. document scope, near-duplicate handling, novelty/diversity). *Pros:* high fidelity to guidelines; strong commonsense/domain reasoning; robust to subtle intent shifts. *Cons:* subjectivity and assessor drift; fatigue reduces consistency; inter-annotator agreement (IAA) can be modest on borderline cases.

**LLM.** LLMs apply instructions consistently at scale and can follow grading rubrics. However, they may rely on surface similarity, underweight collection-specific rules, or miss latent/temporal intent; outputs are sensitive to prompts and calibration. *Pros:* high procedural consistency; scalable graded labeling; fast iteration for rubric refinement. *Cons:* over-reliance on lexical overlap or prior knowledge; length/position biases; stale knowledge; difficulty with multi-intent/implicit queries and source credibility.

### 2. Cost and Speed

**Human.** Expert judging is costly and time-consuming; throughput drops as pool depth and query count increase. *Pros:* reliable gold labels for evaluation and model diagnosis. *Cons:* high monetary/time cost; slower scaling to large corpora.

**LLM.** Batch/API-driven labeling scales to millions of pairs with low marginal cost; suitable for pre-labeling and continuous refresh. *Pros:* low cost per instance; high throughput; easy to expand pools. *Cons:* initial setup (prompt/rubric/tuning) and quality control still required.

### 3. Error Profiles

**Human.** Errors stem from inattentiveness, inconsistent application of guidelines, anchoring, and coder drift. *Examples:* inconsistent use of graded scales; confusion between topical relatedness vs. task utility; overlooking time-sensitivity.

**LLM.** Errors reflect prompt/rubric mismatch, calibration issues, training-data bias/leakage, and granularity confusion (passage vs. document). *Examples:* rewarding lexical overlap despite low utility; penalizing relevant but short snippets; temporal mistakes (outdated facts); inconsistent handling of duplicates/near-duplicates.

## C    EFFECT OF THE NUMBER OF POSITIVE DOCUMENTS

**Setup.** We train a DPR retriever and use $\tau = 0.95$ to select positive documents per query via our PFN procedure with an LLM (Llama 3.1 70B). For each query, let $k$ denote the number of positives actually used for training. We evaluate four settings: $k \in \{1, 2, 3, 3.5\}$, where $k = 3.5$ indicates that *all* documents passing the threshold $\tau = 0.95$ were included; across queries this yields an average of 3.5 positives per query. Performance is reported as Recall@20 on the development set under an otherwise identical training protocol.

**Results.** As shown in Figure 5, we vary the number of positives used for training the DPR under our PFN selection pipeline while keeping all other settings fixed. With the threshold set to $\tau = 0.95$, the PFN selector yields on average 3.5 positives per query. We then train with the *Confidence-guided Multi-Positive (CMP)* loss while increasing $k \in \{1, 2, 3, 3.5\}$, where $k = 3.5$ denotes using *all* documents that pass the $\tau = 0.95$ threshold for each query (the average across queries is 3.5). Recall@20 improves monotonically with $k$: 79.4 ($k$=1), 80.0 ($k$=2), 80.9 ($k$=3), and **81.1** ($k$=3.5); the gains over $k$=1 are +0.6, +1.5, and +1.7 points, respectively. The curve in Figure 5 peaks at $k = 3.5$, indicating that—when trained with CMP—leveraging *all threshold-qualified positives* per query is most effective; CMP can aggregate complementary evidence from multiple positives while remaining robust to residual noise.

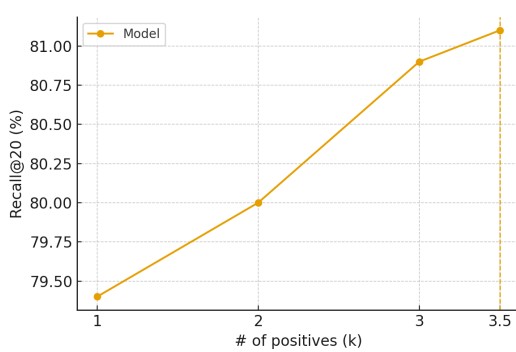

Figure 5: DPR performance (Recall@20) as a function of the number of positives $k$. The vertical dashed line marks the dataset mean $k = 3.5$. LLM for PFN selection: Llama 3.1 70B; threshold $\tau = 0.95$.

