# OpenReview forum: "CLEAR: Consistent Labeling Enhanced by LLM-driven Automated Re-labeling for Improved Information Retrieval"
_ICLR.cc/2026/Conference — Submitted to ICLR 2026_

### Official Review · Reviewer_sp7p · 2025-10-20

**Soundness:** 2
**Presentation:** 2
**Contribution:** 2
**Rating:** 2
**Confidence:** 4

**Summary:**

The paper introduces CLEAR, a framework that utilizes LLMs for training data relabeling. The framework uses LLMs to score the retrieved context, and flip their labels to positive if the score is high. With multiple positive documents, the authors also introduce new training losses for finetuning. The experiments show that the dense retrieval models finetuned on the relabeled data have improved retrieval metrics on 4 datasets.

**Strengths:**

* The paper introduced new loss functions for retrieval tasks with multiple positive documents.
* Various dense retrievers are compared under different settings.

**Weaknesses:**

* The methodology relies on several unjustified details. Please see the following Questions.
* The evaluation results and corresponding analysis are questionable. Please also see the details in Questions.
* The practical application scenario for CLEAR appears limited. The method requires the retriever to already have a strong initial performance; otherwise, "Potential False Negatives" with low similarity scores cannot be identified. Thus, it is problematic to apply the method to pretrained checkpoints. Conversely, if the framework is intended to refine retrievers for specific tasks, the small improvements (as reported in Table 1) may not be worth the costs of LLM annotation and the GPU hours required for finetuning.
* There are a few formatting issues in the paper.

**Questions:**

## Methodology

* In the formula $$
    cs = 1-d(GT, p(y|T, q, d))
    $$
on line 185, the symbol `d` is used to represent both the distance function and documents. To improve clarity, please consider using a different symbol for one of these terms.

* Please clarify the structure of the ground-truth vector `GT`. Line 185 describes it as a binary vector, while line 191 implies it corresponds to a sequence of `L` tokens. Is `GT` a sequence of one-hot encodings? If so, please elaborate on how the full probability distribution
    $$
    p(y|T, q, d)
    $$
    is calculated for the entire token sequence. I can understand how the distribution for the first token is derived, but how about the following ones? Additionally, why choose the L2 distance metric over a more conventional measure like the KL-divergence when comparing distributions?

* Regarding the processes in Sections 2.5.1 and 2.5.2, will a query with a non-empty PFN set always result in at least one document being relabeled as positive by the LLM?

* The proposed method focuses on relabeling negative documents as positive. Could you explain the reasoning for not also relabeling documents from positive to negative? This seems to be a more appropriate correction for the case illustrated in Figure 1.

* What is the justification for treating the LLM-relabeled documents as "true positives"? Is it possible for them to be "false positives"? If so, how would such misclassifications affect the validity of the hypotheses and analysis in Sections 4.2-4.5?

### Evaluation

* The performance gains reported in Table 1 appear marginal in many cases. To substantiate the claim of improvement, please consider reporting confidence intervals or standard errors for the baseline results, possibly calculated via bootstrapping, to demonstrate statistical significance.

* Could you please provide details on the computational cost of the CLEAR method? Specifically, at what stage does the LLM annotation occur: is it a one-time preprocessing step or integrated into the finetuning loop? Furthermore, could you quantify the number of LLM calls required for each retriever?

## Other

* The claim on lines 39-40 that "several widely used benchmark datasets have been shown to contain a non-negligible number of incorrect labels" is a key motivation for this work. Please provide citations or other evidence to support this assertion.

* The in-text citations currently lack parentheses, which can make them difficult to distinguish from the main text. Please use the `\citep` command as specified in the submission template to ensure proper formatting.

* There appears to be a formatting issue with the vertical spacing between sections. For example, on line 97, the section heading is too close to the preceding paragraph. Please review the document layout to ensure adequate spacing throughout.

---

### Official Review · Reviewer_R3o2 · 2025-10-30

**Soundness:** 3
**Presentation:** 3
**Contribution:** 3
**Rating:** 6
**Confidence:** 4

**Summary:**

The paper presents CLEAR, an LLM-based pipeline designed to address the persistent issue of noisy and inconsistent labels in information retrieval (IR). CLEAR employs a structured five-stage process for automated re-labeling and further explores the complementary value of label quality and diversity through multi-positive training strategies. The proposed method is well-motivated and empirically validated across several major IR benchmarks (MS MARCO, Natural Questions, TriviaQA), showing consistent gains over strong baselines. Overall, the work offers a solid contribution toward improving IR systems via data-centric label refinement.

**Strengths:**

1.The paper introduces a novel LLM-driven re-labeling framework for information retrieval, shifting the focus from model-centric to data-centric improvement. Its integration of confidence-based and multi-positive document training is interesting and rarely explored in IR.
2.The proposed five-stage pipeline is methodologically sound, and the authors design comprehensive strategies and experiments to validate the effectiveness of the approach. The experimental results clearly demonstrate the strong performance and robustness of the CLEAR method.
3.CLEAR achieves consistent improvements on major IR benchmarks and strong baselines, confirming the practical value of LLM-based label refinement. The approach has broad potential for future data-centric IR and LLM research.

**Weaknesses:**

1.The computational overhead and cost of the CLEAR pipeline are not reported, leaving its overall scalability and practicality somewhat unclear. A more detailed analysis of efficiency (e.g., batch processing strategies, runtime, or LLM inference cost) would strengthen the discussion of real-world applicability.
2.The thresholds ϕ and τ are critical hyperparameters that strongly influence model performance. However, the paper lacks a principled and detailed explanation, which affects the reproducibility and perceived rigor of the methodology.
3. Experiments are limited to English benchmarks. Evaluating CLEAR on multilingual or domain-specific datasets would help verify its generalizability and robustness across diverse retrieval settings.
4. Table 1 is overly dense and could benefit from clearer segmentation by retriever type or dataset to enhance readability. Adding more informative visualizations or summarizing key results in figures would make the experimental findings easier to interpret and more engaging.
5. The paper provides limited theoretical or analytical insight into why LLM-based re-labeling improves retrieval effectiveness. A brief discussion on the underlying mechanisms could further enrich the contribution.

**Questions:**

According to the experimental results, the reported 2–3% improvement is achieved only when combining human and LLM-generated labels, while the performance gap between purely human-labeled and purely LLM-labeled data remains small. Moreover, the paper indicates that noticeable gains occur mainly with larger LLMs. Given this,
1.	Could the authors clarify whether the overall cost-effectiveness of CLEAR is sufficient?
2.	Specifically, why not invest in improving human annotation quality (e.g., through better training or guideline refinement) instead of relying on computationally expensive LLM-based post-processing?
3.	Could CLEAR be extended to cross-modal retrieval (e.g., image–text pairs)?
4.	Did you explore multi-round CLEAR for further improvement?

---

### Official Review · Reviewer_KxK3 · 2025-10-30

**Soundness:** 2
**Presentation:** 3
**Contribution:** 2
**Rating:** 2
**Confidence:** 5

**Summary:**

The work proposes an LLM-based pipeline to relabel existing labels and find missing positives. Using this pipeline, a better dataset is constructed, and training on it results in better performance.

**Strengths:**

- The paper is mostly clearly written.
- The proposed method shows strong improvement over the included baselines.
- The proposed method is generic, and can be applied on top of any Dense Retriever algorithm.
- Extensive ablations have been conducted.

**Weaknesses:**

- **1. Relevance of Contributions in reference to missing label literature in Extreme Classification:** It would be interesting to know how the contributions made in this work differ from those addressing the missing label problem in the extreme classification literature [1]. I would recommend adding a related work section on extreme classification covering literature on missing labels.

- **2. Section 2.4 is unclear:**  It would really help to (i) include an example showing what the prompt template $T$, the answer $a_i$​, and the ground truth (GT) are for a given query, (ii) clarify whether GT is different from $a_i$​, and (iii) use a different symbol for the distance metric. Currently, I believe ddd is being used to represent both the document and the distance metric.

- **3.Rationale for the distance metric is unclear:** How do you get the probability of GT tokens ($p_h$​ here)? Are these one-hot vectors? I’m also unclear about the rationale for the distance metric in Equation (4). For example, would simply asking the LLM to determine whether qqq and ddd are related given GT be a more robust way to derive a confidence score? It would look something like this:

*Prompt Template*: You will be given a question, a document, and a ground truth answer. Your task is to tell whether the question is relevant (answer in Yes/No) to the document by using the information from the ground truth answer.

Later, you could look at the probability of “Yes” or “No” as a measure of confidence. The reason I find the current metric unsatisfying is that, depending on the LLM, you could have different values of per-token probabilities depending on the ground truth. Two GTs could convey the same semantic meaning but be worded very differently, resulting in very different values for the distance metric.

- **4.Novelty of the AMP loss:** I believe that variants of the contrastive loss with multiple positives have already been discussed in past works such as [2].

- **5. Missing Baselines:** From what I understand, the experimental setup uses five different dense retrievers and shows that the proposed relabeling scheme improves performance across all of them. While the improvements are consistent, I believe there are relevant classes of baselines such as LLM-based augmentation [3]. It would be interesting to see if the proposed relabeling scheme outperforms works like [3]. Note: I understand that doing this within the rebuttal timeframe is challenging, but including it in a subsequent version of the paper would help make the method more convincing.

- **6. Ability of the pipeline to recover missing labels:** I believe the TREC-DL 19 and TREC-DL 20 datasets come with extensively annotated labels, where each query has multiple positives. My hunch is that many relevant labels may be omitted in the top-K filter from Stage 3. To test this, I recommend that the authors use one of the retrievers trained on the MS MARCO dataset (as the TREC datasets are very similar in nature to MS MARCO) and evaluate what proportion of the true positives from the TREC dataset are recovered by their pipeline.

**Nits**

1. Spacing above “Section 2: Method” seems to be too less?
2. In equation (7) why would the LLM labelled dataset have less than $m$ questions?

**References**

[1]: Prakash, Jatin, et al. "On the Necessity of World Knowledge for Mitigating Missing Labels in Extreme Classification." Proceedings of the 31st ACM SIGKDD Conference on Knowledge Discovery and Data Mining V. 1. 2025.

[2]: Khosla, Prannay, et al. "Supervised contrastive learning." Advances in neural information processing systems 33 (2020): 18661-18673.

[3]: Saad-Falcon, Jon, et al. "Udapdr: Unsupervised domain adaptation via llm prompting and distillation of rerankers." arXiv preprint arXiv:2303.00807 (2023).

**Questions:**

Please see weaknesses above.

---

### Official Review · Reviewer_TyQR · 2025-10-31

**Soundness:** 2
**Presentation:** 2
**Contribution:** 1
**Rating:** 2
**Confidence:** 4

**Summary:**

The paper shows that existing supervised training datasets contain incorrect training labels. They proposed a CLEAR framework that rectifies wrong labels using an LLM and uses those as positives to retrain multiple dense retrieval models.

**Strengths:**

The paper works on a crucial problem of incorrect labels in training datasets. They work with multiple dense retrieval models and evaluate on three datasets, although the setting is in-domain with well-defined (train/dev/test) splits.

**Weaknesses:**

There are many weaknesses and loopholes associated with the paper; here I write them below:

1. **Line 053 claims the CLEAR pipeline is novel**: However, neither the pipeline nor the methodology is novel, and a thorough literature survey is missing. Upon searching, I easily found [1] that proposes a similar idea by using a cascading LLM judge to relabel hard negatives as positives within existing training datasets. In addition, [2] and [3] worked on LLM for relabeling training datasets. Since the paper does not include a related work section, I recommend that the authors conduct a thorough literature review and include and cite related works. In addition, they mention information retrieval in their title, but few related works are mentioned. The paper is missing an LLM judge-related work section and dense retrieval models.

- [1] Hard Negatives, Hard Lessons: Revisiting Training Data Quality for Robust Information Retrieval with LLMs. Thakur et al. EMNLP 2025 Findings.
- [2] Gecko: Versatile Text Embeddings Distilled from Large Language Models. Lee et al. 2024.
- [3] Gemini Embedding: Generalizable Embeddings from Gemini. Lee et al. 2025.

2. **Many crucial details are missing from the paper**: (a) Did you retrain all the models: BGE, GTE, CoCondensor? That is not possible as the code to reproduce and fine-tune GTE and BGE is not available in open source. So, clarifications on how you trained the models are required. Did you take the existing fine-tuned backbones and fine-tune them again on each dataset? This is not advised, as many of these models have already been fine-tuned on MS MARCO, NQ, and other datasets. (b) Ground-truth answer for datasets: MS MARCO does not contain any ground-truth answer within its training pairs. How did you find them or generate them? How did you generate the confidence score for MS MARCO?

3. **The paper focuses on cleaning training datasets, but no information about the dataset or analysis on the data is provided:** How many training pairs are used for relabeling, and what are the statistics after labeling? What percentage of passages do the human and LLM label overlap? The paper is missing an analysis section, which will help the readers to understand how unclean the existing training datasets are.

4. **Top negatives only mined from a single retriever and missing baselines**: In practice, multiple negatives are mined from a variety of retrieval models [4]. The paper only uses the retriever model itself to mine hard negatives, which is not recommended in practice. Many crucial baselines are missing: Instead of relabeling hard negatives, they should compare against techniques that carefully (a) avoid top-k hard negatives according to a top-k threshold [4] and (b) use a cross-encoder to relabel the training pairs, which are cost-efficient in comparison to the LLM used in the paper [5]. I would encourage the authors to compare against suggested techniques and include them as baselines.

- [4] NV-Retriever: Improving text embedding models with effective hard-negative mining. Moreira et al. 2024.
- [5] RocketQA: An Optimized Training Approach to Dense Passage Retrieval for Open-Domain Question Answering. Qu et al. NAACL 2021.

5. **The loss functions suggested in the paper are minor modifications:** They average the InfoNCE loss (in Averaging multi-positive (AMP) loss) and assign dynamic weights in Confidence-guided multi-positive (CMP). Both are minor modifications in the loss function. In addition, during training, it is unclear why Pos=1 is kept in the InfoNCE loss function. In practice, usually multiple positives are present in the training loss function, and one positive passage is chosen randomly during training.

6. **The writing is weak, with many grammatical mistakes:** (1) Figure 1 is missing a full stop (.) at the end of the caption. (2) Method section header formatting is incorrect. I would recommend reviewing the grammar.

**Questions:**

Check weaknesses for my questions.

---

### Meta-Review · Area_Chair_qDJQ · 2025-12-27

**Summary:**

No rebuttal is posted and thus recommend rejection based on the negative initial scores with three rejects except for one borderline accept.

**Reviewer Concerns:**

No rebuttal is posted and thus recommend rejection based on the negative initial scores with three rejects except for one borderline accept.

**Reviewer Scores:**

No rebuttal is posted and thus recommend rejection based on the negative initial scores with three rejects except for one borderline accept.

---

### Decision · Program_Chairs · 2026-01-26

Reject